# Exploration on the Enhancement of Detoxification Ability of Zearalenone and Its Degradation Products of *Aspergillus niger* FS10 under Directional Stress of Zearalenone

**DOI:** 10.3390/toxins13100720

**Published:** 2021-10-12

**Authors:** Jian Ji, Jian Yu, Yang Yang, Xiao Yuan, Jia Yang, Yinzhi Zhang, Jiadi Sun, Xiulan Sun

**Affiliations:** 1State Key Laboratory of Food Science and Technology, School of Food Science and Technology of Jiangnan University, Wuxi 214122, China; jijian@jiangnan.edu.cn (J.J.); jiangsuyzyj@163.com (J.Y.); yinzhizhang@jiangnan.edu.cn (Y.Z.); sunjiadi@jiangnan.edu.cn (J.S.); 2College of Food Science and Pharmacy, Xinjiang Agricultural University, Urumqi 830052, China; yyang7858@163.com; 3Guangzhou GRG Metrology and Test Co., Ltd., Guangzhou 510630, China; yuanxiao@grgtest.com; 4Yangzhou Center for Food and Drug Control, Yangzhou 225000, China; jiajia82112001@163.com

**Keywords:** zearalenone, *Aspergillus niger*, biodegradation methods, degradation products, food safety

## Abstract

Zearalenone (ZEN) is one of the most common mycotoxin contaminants in food. For food safety, an efficient and environmental-friendly approach to ZEN degradation is significant. In this study, an *Aspergillus niger* strain, FS10, was stimulated with 1.0 μg/mL ZEN for 24 h, repeating 5 times to obtain a stressed strain, Zearalenone-Stressed-FS10 (ZEN-S-FS10), with high degradation efficiency. The results show that the degradation rate of ZEN-S-FS10 to ZEN can be stabilized above 95%. Through metabolomics analysis of the metabolome difference of FS10 before and after ZEN stimulation, it was found that the change of metabolic profile may be the main reason for the increase in the degradation rate of ZEN. The optimization results of degradation conditions of ZEN-S-FS10 show that the degradation efficiency is the highest with a concentration of 10^4^ CFU/mL and a period of 28 h. Finally, we analyzed the degradation products by UPLC-q-TOF, which shows that ZEN was degraded into two low-toxicity products: C_18_H_22_O_8_S (Zearalenone 4-sulfate) and C_18_H_22_O_5_ ((E)-Zearalenone). This provides a wide range of possibilities for the industrial application of this strain.

## 1. Introduction

Zearalenone (ZEN) is a kind of Fusarium mycotoxin with a high contamination range and estrogenic activity. It results in considerable economic losses to the food and feed industries, in addition to its accumulation in animals or human bodies via the food chain, causing acute and chronic poisoning of animals or people [1,2]. These common soil fungi are found in temperate and warm regions and often contaminate cereal crops, such as barley, corn, oats, rice, and sorghum [3,4]. 

ZEN poisoning can be categorized into acute poisoning and chronic poisoning. In acute poisoning, the animals show restless excitement, staggering, muscle tremor, and sudden death [5]. Cyanosis of the mucous membrane and the absence of an apparent change in body temperature are also present [6]. In chronic poisoning, harmful effects are mainly observed in female animals, one of which is the swelling of the female genitals [7]. 

ZEN contamination in feed occurs worldwide. Africa has become an area with a high incidence of mycotoxin pollution due to climatic reasons, poor sanitation and epidemic prevention conditions, and lack of sufficient food safety knowledge among the people [8]. In Europe, the pollution of mycotoxins is also serious. Streit [9] conducted a joint test of multiple mycotoxins on 17,316 feed and feed ingredient samples from Europe. It was found that 72% of the samples were found to have at least one type of mycotoxin contamination, and 38% of the samples were found to be contaminated with multiple mycotoxins [9]. In addition, there are also reports of a certain amount of mycotoxin contamination in food and dairy products in the United States [10].

ZEN degradation methods are classified as have been identified: physical, chemical, and biological methods. Physical methods mainly include ultraviolet radiation, heat treatment, and X-ray treatment [11], but these physical techniques have not been widely used because of their low degradation rate and possible damage to food. Commonly used chemical methods include ozone treatment [12], reducing agents, and alkali treatment. However, chemical methods are difficult for food safety organizations to adopt because chemical degradation agents are retained in food, and mycotoxins have low degradation efficiency.

In contrast, the biodegradation method to degrade mycotoxins has the characteristics of better targeting and high degradation rate. The current methods of biodegradation of mycotoxins mainly originate from four aspects: bacterial degradation (*Bacillus subtilis* [13], etc.), fungal degradation (*Gliocladium rosea* [14], etc.), enzyme degradation (ZHD101 [15], etc.), and genetic engineering to select disease-resistant plants [16]. In addition, the biodegradation method can further prevent the human health threats caused by the chemical residues, in turn caused by the chemical degradation method and the destruction of food nutrients and texture caused by the physical degradation method [17]. Last but not least, the method of breeding disease-resistant plants through genetic engineering can fundamentally solve the infection problem of toxin-producing fungi. Compared with physical and chemical methods, this has achieved a transformation in thinking from passive degradation of mycotoxins to active prevention and control [17,18]. Therefore, the biodegradation method has very broad development prospects.

Previous studies have reported that FS10 can degrade ZEN and AFB_1_ [19,20]. Prompted by these findings, we obtained the Zearalenone-Stressed-FS10 strain ZEN-S-FS10, which can be used to efficiently degrade ZEN. The strain was deposited in the China General Microbiological Culture Collection Center (CGMCC No. 20745). We also determined the degradation ability, apparent characteristics, and degradation mechanism analysis [21]. Compared with FS10, the removal rate of ZEN by ZEN-S-FS10 has been greatly improved, and the required removal time has also been greatly shortened. This study can provide new ideas and insights for efficient and rapid ZEN degradation.

## 2. Result and Discussion

### 2.1. Analysis of the Strain Information and Degradation Rate

The 18s rDNA sequencing strategy was utilized for genome detection of ZEN-S-FS10, and the sequencing result was in Appendix A. By analyzing the sequencing results and comparing with the literature [22], we found that gene *RsaI* was matched in the sequencing results. According to previous literature reports, this gene *RsaI* can be used to distinguish *Aspergillus niger* from *Aspergillus* section *nigri*, which proved that ZEN-S-FS10 is still *Aspergillus niger* strain. In addition, we analyzed the purity of the bases in the sequencing (Appendix A), which shows that each base presents a single peak shape. This indicates that this strain belongs to a single pure *Aspergillus niger*, instead of another species of the *Aspergillus niger* aggregate.

By analyzing the degradation rate of *Aspergillus niger* FS10 under different stress conditions, we found that at 0–24 h, the degradation rate increased with the increase of stress time, and as the stress time is 24 h, the degradation rate no longer changes significantly (Figure 1A). When the ZEN concentration used during stress is 1.0 μg/mL, the degradation rate can also be maintained at a high and stable level. Moreover, due to the high ZEN concentration at 10 μg/mL, *Aspergillus niger* was difficult to grow, so the degradation rate decreased significantly (Figure 1B). In addition, as the stress times increased, the degradation rate also increased significantly and finally reached stability at G5 to G6 (Figure 1C). Therefore, the optimal stress condition is that, 1.0 μg/mL ZEN is used for 24 h, and stress times is 5. The ZEN stressed FS10 strain was named as ZEN-S-FS10. The ZEN degradation rates of FS10 with different stress time was in Appendix A.

HPLC was utilized for degradation rate analysis of ZEN, with recovery rat exceeded 90% (Appendix A). As shown in Figure 1C, the average degradation rate of ZEN by the original strain is 83.51% [23], and the average degradation rate of ZEN-S-FS10 (G5) reaches 96.00%. As the number of stresses increases, degradation rate has increased significantly from G3 to G5. The hypothesis for this phenomenon is that the ZEN stress leads to the changes in the metabolism of FS10, like producing corresponding enzyme substances for self-protection [24]. This change in the metabolism, resulting in changes in enzyme levels, may be the main reason for the increased degradation of ZEN.

### 2.2. Effect of Stress on the Myceliume and Growth of Myceliume

Using SEM, we analyzed the surface characteristics of the mycelia after stressed. More convex lines appeared on the surface of the stressed mycelium than that of the FS10 (Figure 2A,B). In the same dry state, the stressed mycelium was more prone to water loss, compared with the original strain, which may be attributed to ZEN stimulation. Under the stress environment of ZEN, the mycelium needs to synthesize more degrading enzymes for self-protection. This process may reduce the ability of the mycelium to retain water [25,26], which leads to the surface tissue of the mycelium considerably shrinking, potentially producing more protuberant lines.

We compared the dry weight changes in the mycelium of FS10 and ZEN-S-FS10 as degradation time increased (Figure 2C). It was found that with the increase of the cultivation time, the dry weight of the two myceliums increased and stabilized with time. We also found that when stability is reached, the dry weight of ZEN-S-FS10 mycelium will be less than that of FS10. This phenomenon indicates that as the stress of ZEN to *Aspergillus niger* increases, it will further inhibit the growth of mycelium, but the degradation rate of ZEN by *Aspergillus niger* has been significantly improved. The reason might be that ZEN-S-FS10 is stimulated by ZEN in the spore state, leading to changes in metabolism, which inhibits the metabolic process related to mycelial growth and cannot maintain the original mycelium weight [27,28]. However, FS10 has not been stimulated or inhibited by any ZEN in the spore state; thus, the growth volume is better than that of ZEN-S-FS10.

### 2.3. Mycelial Metabolomics Analysis

To demonstrate that the metabolism of the ZEN-S-FS10 changed after directional stress of ZEN, we compared the change in metabolic profile. As shown in Figure 3A, the metabolites of total ion chromatogram (TIC) of FS10 before and after ZEN stress is not much different, but it can be observed that there are differences in the TIC map at about 6, 9, and 14 min, which indicated that ZEN stress could lead to the metabolism change, consistent with the previous hypothetical results on the difference in degradation rate and water holding capacity.

It could be noticed in Figure 3B that the PCA plots of FS10 and ZEN-S-FS10 metabolites are separated, which proves that differences in metabolism exist between the FS10 and ZEN-S-FS10. The volcano plot (Figure 3C) shows the differences in metabolites between FS10 and ZEN-S-FS10, indicating that the metabolites in the ZEN-S-FS10 strain exhibited an upregulated trend. These upregulated metabolites are mainly related to the metabolism of amino acids and organic acids, such as DL-aspartic acid, 3-Methyl-2-oxobutanoic acid. The increased metabolism of these small molecules may be attributed to the stress response or self-protection mechanism of FS10 under the stress of ZEN. Moreover, referring to the metabolic pathways in Figure 3D, we analyzed all metabolic pathways related to it. The results show that the contents of metabolites such as 2-oxoglutarate, oxalacetic acid, and L-asparagine, among others, were upregulated, and the content increased significantly. These substances related to metabolic pathways, such as the TCA cycle, may affect the ZEN degradation ability. Therefore, this point can indicate that the targeted stimulation produced a more obvious change in the internal metabolic pathway of ZEN-S-FS10. Previous studies have shown that the metabolism of amino acids is very important for biological cells to cope with the adverse external environment [29]. Amino acid metabolism can quickly secrete substances that protect cell tissues to reduce the adverse effects of the external adverse environment on cells [30,31]. The improvement of these physiological indicators may largely affect the degradation ability of ZEN-S-FS10.

### 2.4. Analysis of the Optimization of Degradation Conditions

To explore the best degradation conditions for ZEN-S-FS10, we set the strain concentrations to low, medium, and high levels. The degradation effect for each time period was detected. As shown in Figure 4A, the degradation rate of the high-concentration (10^5^ CFU/mL) and medium-concentration (10^4^ CFU/mL) strains exceeded 97%, whereas that of the low-concentration strain (10^3^ CFU/ mL) was only about 71%. However, the degradation rate of the medium concentration strain (10^4^ CFU/mL) to reach the optimal degradation rate was about 4 h shorter than that of the high concentration strain (10^5^ CFU/mL). The optimal degradation time is 28 h, and the optimal degradation concentration is 10^4^ CFU/mL.

### 2.5. Kinetic Study of ZEN Degradation

To explore the degradation process, the reaction kinetics should be determined. The first-order kinetics of degradation was studied by logarithmic transformation. The first-order reaction kinetics can be expressed using (1):(1)$$ln\frac{C_{0}}{C_{t}}=k_{1}t$$
where *C*_0_ and *C_t_* represent the residual concentrations of ZEN in the initial state and the stable state, respectively, and *k*_1_ (h^−1^) is the first-order kinetic constant. Kinetic analysis was conducted with the strain set to medium concentration: the strain concentration was set to 10^4^ CFU/mL, and the degradation time was 28 h. During the first 28 h of degradation, ZEN concentration changed from 0.49354 μg/mL to 0.01161 μg/mL, and *k*_1_ ranged from 0.003251 h^−1^ to 0.06385 h^−1^. The results are listed in Figure 4B and Appendix A. The functional relationship between *k*_1_ and concentration (*C*) is given by (2):(2)$$k_{1}=0.0002C+0.077$$

After fitting, R^2^ = 0.9846 > 0.9. The degradation curve for the *Aspergillus niger* ZEN-S-FS10 was in accordance with the first-order degradation kinetic model [32]. Therefore, the first-order degradation kinetic model can be used to analyze the degradation process.

### 2.6. Stability of Strain Degradability Analysis

As shown in Figure 5A, the ZEN degradation rate of ZEN-S-FS10 exceeded 95%. Fungi will have an increase in the degradation ability of ZEN after ZEN stress, but the degradation ability may gradually decrease with the increase of the number of passages. However, the degradation rate of ZEN-S-FS10 showed no significant decrease after subculture. Thus, owing to the stability of the ZEN degradation capability in ZEN-S-FS10, this increase in the degradation of ZEN caused by metabolic changes is a stable change.

### 2.7. Analysis of the Influence of Fermentation Environment Changes in Degradation Ability

EDTA, Na^+^, Cd^2+^, alkali (pH = 11.0), and acid (pH = 2.0) were chosen as influencing factors to evaluate the ZEN degradation efficiency of ZEN-S-FS10. The result in Figure 5B shows that Na^+^ has a minimal effect on the degradation rate, and other influencing factors exert more obvious effects on the degradation rate. Among them, acid exerts the greatest effect on the ZEN degradation ability of ZEN-S-FS10 (60%). Therefore, ZEN-S-FS10 needs to maintain a good degradation rate of ZEN when the environmental pH is neutral. Moreover, previous studies have shown that changes in pH can affect the fermentation process of microorganisms [33,34]. To achieve the best degradation effect of ZEN-S-FS10, avoid degradation in an over-acid and over-alkali environment.

### 2.8. Analysis of the Degradation Site of the Mycelium

In order to further analyze whether the degrading enzyme secreted by the mycelium is an extracellular enzyme, we measured the degradation rate of the fermentation broth after inactivation. As shown in Figure 6, by comparing the degradation ability of different parts at different time points, it can be seen that the degradation ability of the fermentation broth is significantly stronger than that of other parts. Moreover, the ZEN degradation ability of the fermentation broth was mostly lost after inactivation, as determined by measuring the degradation ability of the inactivated fermentation broth. The ZEN degradation efficiencies in descending order were as follows: fermentation broth > mycelium > spore > inactivated fermentation broth. Therefore, an extracellular enzyme secreted by the mycelium plays the most important role in degrading ZEN [35]. The enzyme also cannot tolerate high temperatures. Previous studies have shown that microorganisms can secrete extracellular enzymes for mycotoxin degradation or dealing with other adverse conditions [36].

### 2.9. Analysis of Probable Degradation Products and Degradation Pathways

Since it has been proved that the main degradation site of ZEN-S-FS10 is the extracellular enzyme secreted from the hyphae into the fermentation broth, the research mainly focused on the fermentation broth. By comparing the UPLC-q-TOF data of the blank group, the fermentation broth group, and the crude protein enzyme solution group after ultrafiltration, two possible ZEN degradation products were identified: product A: C_18_H_22_O_8_S (Zearalenone 4-sulfate) and product B: C_18_H_22_O_5_ ((E)-Zearalenone).

The elemental composition of product A was C_18_H_22_O_8_S, *m*/*z* = 397.09671, which was detected at RT = 5.617 min. This product was calculated using the MSFINDER 3.46 software, with accuracy score = 6.95. Figure 7A shows the changes in the content of this degradation product in the 3 groups of samples. The content in the blank control group was considerably low, with almost no such substance, and the content in the fermentation broth after degradation was significantly increased. In the purified crude enzyme solution, the content was higher, but owing to a lack of nutrients, the content was slightly lower than that in the degraded fermentation broth. These differences prove that the product was produced after the degradation of *Aspergillus niger* ZEN-S-FS10.

Figure 7B shows the fragment ion peaks of this product. The MSFINDER 3.46 software was utilized for the elemental composition and structure elucidation of product, and the product was annotated as C_18_H_22_O_8_S (Zearalenone 4-sulfate), based on the accurate mass of precursor ion and rules of hydrogen rearrangement.

Generating this kind of product probably requires that the hydroxyl group of C4 and the S atom in the degrading enzyme provided by ZEN-S-FS10 undergo a sulfhydrylation reaction, and the original hydroxyl group be oxidized to form a sulfate group. Studies show that enzymes secreted by certain microorganisms can undergo such sulfhydrylation reactions [37]. This degradation product is reported in the literatures [38,39], which indicate that after food products contaminated with ZEN and T-2 toxins are properly detoxified, the Zearalenone 4-sulfate is extracted [40]. These findings, together with the results of this experiment, indicate that the degradation product is one of the products produced after ZEN degradation. Studies also show that after ZEN undergoes sulfhydrylation, the toxicity is greatly reduced relative to the original structure even though the toxicity of Zearalenone 4-sulfate is not completely lost [41].

As shown in Figure 7C, the change in content was highly similar to the change in product A. This substance did not exist without degradation, and its content was substantially increased after degradation. Therefore, this substance could only be produced after the degradation of ZEN. As shown in Figure 7D, the fragment ion peak of product B is obtained. The structural accuracy score of product B reached 6.34, and the MS1 tolerance is 0.5474 mDa. With the analysis of MSFINDER 3.46, the product B was annotated as C_18_H_22_O_5_ ((E)-Zearalenone). Product B and the original structure of ZEN only slightly varied. The difference was only observed in the conformational position of the methyl group and the H atom at C_11_. Compared with that of the original ZEN molecular structure, the spatial positions of the methyl group and H atom were switched around, forming the isomers of the ZEN molecule. In Appendix A, the spatial structure of product B is compared with that of the ZEN molecule. Isomers with the same elemental composition exhibit different properties because of variations in spatial structure, which may decrease toxicity [42]. However, in the current study, the toxicity of product B remained unclear, and its toxicity needed to be confirmed.

The content peak intensities of the two products in the samples after degradation were compared. According to the reduction of ZEN molecules, about 30% of the degradation was converted to product A, whereas only less than 1% was degraded into product B (Figure 7E,F). No other changes in product content were consistent with the changes in these two products. We thus speculated that product A was the main degradation product, and other products could be degraded into a large number of small molecules. Earlier research [43] indicates that the toxic structure of mycotoxins often exists in the epoxy structure of toxins. Therefore, when the epoxy structure of the toxin is broken or destroyed, the toxicity of mycotoxins disappears or decreases. Therefore, *Aspergillus niger* ZEN-S-FS10 cannot completely destroy the ZEN structure, but it can greatly reduce its toxicity by changing the functional group attached to the ZEN molecule and changing the position of some atoms.

## 3. Conclusions and Prospect

On the basis of the ZEN degradation efficiency of FS10, we used ZEN to stress the FS10 for 5 times. ZEN-S-FS10, a stressed strain with high ZEN degradation efficiency, was ultimately obtained, with a ZEN degradation rate exceeding 95%. The 18s rDNA and base sequence alignment of this strain confirmed that it is still *Aspergillus niger* [22]. In addition, we used metabolomics study to prove that metabolic differences were observed between ZEN-S-FS10 and FS10, and the main differences in amino acid metabolism could affect the ZEN-degradation efficiency. The optimal degradation conditions were determined: a strain concentration of 10^4^ CFU/mL and degradation time of 28 h. We further examined the changes in the physiological indexes of the strain after mutation and found that the mutation influenced the physiological indexes of the strain. Moreover, we analyzed the process of ZEN degradation by ZEN-S-FS10. When the strain was degraded in the PDB medium, it secreted an extracellular enzyme that was not heat-resistant to decompose ZEN. Finally, we used UPLC-q-TOF to analyze 2 degradation products of ZEN: C_18_H_22_O_8_S, with an *m*/*z* of 397.09671 (Zearalenone 4-sulfate) [44] and C_18_H_22_O_5_ with an *m*/*z* of 317.14139 ((E)-Zearalenone) [45].

However, the enzyme class used to degrade ZEN in this strain has yet to be determined [46]. Moreover, *Aspergillus niger* cannot be directly used in ZEN degradation in food. The aforementioned extracellular degradation enzymes have to be further extracted and purified to develop products with high ZEN degradation efficiency [1,47]. From this perspective, metabolism-enhanced FS10 *Aspergillus niger* ZEN-S-FS10 exhibits research and application potential, which bears significance for ZEN degradation in the future [48,49]. The research ideas and development methods related to this strain are expected to prompt efforts toward the degradation of other mycotoxins.

## 4. Materials and Methods

### 4.1. FS10 and Chemicals

*Aspergillus niger* FS10, a non-toxigenic filamentous fungus isolated from fermented Chinese soybean, was acquired from the China Center for Culture Collection (CCTCC NO: M2013703) [19]. The culture methods and conditions applied were consistent with those presented in previous reports [19,50]. ZEN (analytical standard, purity ≥ 99%), which was purchased from Enzo Life Sciences, Inc. (Beijing, China).

### 4.2. Coercion Method and Condition Optimization

A spore suspension was obtained by eluting *Aspergillus niger* FS10 spores from the Potato Dextrose Agar (PDA) medium with 0.9% saline containing 0.05% Tween (China National Pharmaceutical Group, Shanghai, China). The spore suspension was diluted with 0.9% saline containing 0.05% Tween. The concentration of the spore suspension was maintained at 10^6^ CFU /mL. As much as 10 mL of the spore suspension was added to the ZEN solution, and the concentration gradient of ZEN is selected as 0, 0.5, 1.0, 2.0, and 10.0 μg/mL [50]. The spore suspension containing ZEN was cultured in a shaker. The cultures were incubated at 28 °C and 180 rpm for 0, 12, 24, 36, and 48 h. Exactly 1 mL of the mutated spore solution was dripped into the PDA medium for subculture and then placed in a fungal incubator at 28 °C for 5 d. This process was conducted from 1 to 6 times. The fifth-generation stressed strain served as the target strain [50]. By comparing the degradation rate of ZEN under different stimulation conditions, the optimal stimulation conditions were finally determined, and the target strain ZEN-S-FS10 was obtained. The specific cultivation process is shown in Figure 8. At the same time, we used the gene identification method of *Aspergillus niger* and other *Aspergillus* provided by Amaia [22] to sequence the 18s rDNA strain in Wuhan Baiyihui Energy Biotechnology Co., Ltd., and obtained a single band. The sequencing results were posted on the NCBI website (https://blast.ncbi.nlm.nih.gov/Blast.cgi, accessed on 22 September 2021) to conduct comparative analysis to identify the species of the samples submitted for inspection. At the same time, it was compared with the key genes of *Aspergillus niger* provided by Amaia [22] to determine the specific species name. In addition, we tested the peak intensity of the base sequence obtained by sequencing to determine the purity of this strain of *Aspergillus niger*.

### 4.3. Evaluation of the ZEN Degradation Rate

The spore suspension was added into the control Potato Dextrose Broth (PDB) medium and a PDB medium containing ZEN. The ZEN concentration in the PDB medium was controlled to 0.5 μg/mL after detection by liquid chromatography. After sampling the spore suspension, counting on a hemocytometer was used to calculate the spore concentration, and the spore concentration was controlled by the method of diluting with normal saline, and the spore suspension in the PDB medium was controlled to 10^6^ CFU/mL. The spore suspension containing ZEN was cultured in a shaker. The cultures were incubated at 28 °C and 180 rpm for 48 h [51]. After 48 h, 1 mL of the PDB fermentation broth was removed from each group of samples. For each group, 3 parallel samples were measured (*n* = 3). Chloroform was used to extract the sample twice, and 3 mL of chloroform was added each time. A total of 6 mL of the extract was obtained from each sample. The extract was placed in a vacuum freeze dryer for 2 h. The sample was then redissolved with 1 mL of acetonitrile:water = 1:1 solution, and vortex-induced vibration was conducted for 30 s. The samples were ultimately filtered with a 0.22 μm water filter membrane and packed into a chromatographic bottle. The samples were then refrigerated at 4 °C [52].

The ZEN content in the sample was determined by high-performance liquid chromatography (HPLC) using an Agilent 1260 Infinity under the following conditions: C_18_-column, 150 mm × 4.6 mm i.d.,4 mm particle size. Mobile phase conditions: A, methanol (8%); B, acetonitrile (46%); and C, ultrapure water (46%). The mobile phase velocity was 1.0 mL/min, the excitation light wavelength was 274 nm, and the emission light wavelength was 440 nm. The injection volume was 20 μL, and the column temperature was 30 °C [53].

### 4.4. Effect of Stress on the Surface Structure and Growth of the Mycelium

The mycelia of FS10 and ZEN-S-FS10 were examined. The dry weight and surface characteristics of the mycelia after stressing were tested. The filter paper *m*_1_ was weighed before suction filtration. The suspended mycelium in the culture medium was completely poured out for suction filtration treatment. The filter paper and mycelium were placed in the oven to remove moisture. Drying was conducted at 55 °C for 24 h [54]. Subsequently, the filter paper and mycelium were removed and then weighed to achieve the total weight *m*_2_ of the mycelium and filter paper [55]. The mycelial dry weight *m*_3_ was calculated as (3):*m*_3_ = *m*_2_ − *m*_1_
(3)

About 0.1 g of dried mycelium was visualized by scanning electron microscopy (SEM), which was conducted under the following conditions were as follows: SU8020, 3.0 kV, 8.4 mm × 20.0 k [56].

### 4.5. Metabolome Analysis of Strains

To further explore the in-depth changes in ZEN-S-FS10 compared with those in FS10, metabonomic analysis was performed on the hyphae of ZEN-S-FS10. The specific sample procedure preparation methods for metabolome analysis were as follows: A sample weighing 50 mg (±2%) was poured into a 2 mL centrifuge tube. A 500 μL mixed solution consisting of acetonitrile: isopropanol: water (3:3:2, *v*/*v*/*v*) and 3 steel balls were then added to the sample. A high-throughput tissue grinder was used to vibrate for 10 min. After completion, an ultrasound was conducted at 25 °C for 5 min. A 500 μL mixed solution of acetonitrile: isopropanol: water (3:3:2, *v*/*v*/*v*) was again added, and ultrasound was performed at 25 °C for 5 min. The samples were placed in a high-speed centrifuge, which was operated at 14,000 rpm, 4 °C for 5 min. Subsequently, 500 μL of the supernatant was taken from each sample, transferred into a new 2 mL Centrifuge tube, placed in a vacuum freeze dryer, and volatilized to dryness. Up to 80 μL of the 20 mg/mL methoxyamine solution was added to the thoroughly evaporated sample to reconstitute, vortex, and shake for 30 s. The mixture was incubated at 60 °C for 60 min, and 100 μL trifluoroacetamide was added. The obtained mixture was again incubated at 70 °C for 90 min and centrifuged. Up to 100 μL of the supernatant was taken and then added to the liner in GC-MS detection bottles. Each sample had 3 biological parallels [50].

### 4.6. Optimization of Degradation Conditions

To determine the best degradation duration, we evaluated the degradation rate at each time point. The spore suspension of the target strain was prepared with 0.9% saline and then added to the PDB medium. The spore concentration in the medium was controlled to 10^5^ (high), 10^4^ (medium), and 10^3^ (low) CFU/mL. PDB culture method as above was used. The ZEN content in the PDB medium was detected every 4 h [57].

### 4.7. Stability of Strain Degradability

To ensure that ZEN-S-FS10 maintains a relatively stable degradation ability, we tested the ability of ZEN-S-FS10 to degrade ZEN after multiple passages. The spores of ZEN-S-FS10 were prepared into a spore suspension with 0.9% saline containing 0.05% Tween. The spore concentration was maintained at 10^6^ CFU/mL, and 1 mL of the aforementioned spore suspension was added to the PDA medium for a subculture [51]. The same PDA training method as above was applied. Five generations of strain were obtained by cycling the aforementioned process. A spore suspension with 0.9% saline was prepared using the spores of each generation. A spore suspension was incorporated into the control PDB medium containing ZEN. The ZEN concentration in the PDB medium was controlled to 0.5 μg/mL. The spore suspension in the PDB medium was controlled to 10^6^ CFU/mL. The same PDB culture method as above was used for 48 h. After 48 h, 1 mL of the PDB fermentation broth was taken from each group of samples. For each group, 3 parallel samples were measured (*n* = 3). The extraction and detection techniques earlier described were used to detect the ZEN degradation rate.

### 4.8. Influence of Fermentation Environment on Degradation Ability

The spores of ZEN-S-FS10 were cultivated in the PDB medium containing 0.5 μg/mL ZEN by using the previously described method. Each group of PDB was limited to contain 0.1 mol/mL EDTA and NA^+^, Cd^2+^. Two groups—NaOH and glacial acetic acid—were used to maintain the pH of the PDB medium at 11.0 and 2.0, respectively, to achieve the alkaline and acidic conditions for the PDB medium. The samples were incubated at 28 °C at 180 rpm for 48 h. Meanwhile, the ZEN concentration in the PDB was detected by high-performance liquid chromatography to determine the degradation rate. Each group had three biological systems in parallel experiments (*n* = 3) [54].

### 4.9. Study on the Degradation Site

To explore the degradation of ZEN by the ZEN-S-FS10, we evaluated the degradation ability of spores, mycelia, and fermentation broth after mycelium growth. The culture conditions were the same as those previously mentioned. The mycelia cultured in the PDB medium were filtered out completely, washed with sterile water 3 times, and cultured in PBS. To evaluate the degradation rate of the ZEN-degrading enzyme released from the mycelium into the PDB medium, the degradation ability of the inactivated fermentation broth was determined. Half of the PDB medium was added into the sterile centrifuge tube and then inactivated for 10 min in a water bath at 100 °C. ZEN was added into the spore suspension, mycelium suspension, fermentation broth, and inactivated fermentation broth, sequentially; the concentration was then kept at 0.5 μg/mL. The samples were incubated in a shaker at 28 °C and 180 rpm for 48 h. The ZEN degradation rate was detected after culture. The same detection method as earlier described was adopted.

### 4.10. Evaluation of ZEN Degradation Products

To study the degradation products of ZEN-S-FS10 to ZEN, we used reversed-phase liquid chromatography coupled with quadrupole time-of-flight mass spectrometry (UPLC-q-TOF). After using the PDB medium culture ZEN-S-FS10 with a concentration of 10^6^ CFU/mL spore suspension for 48 h, 3 sets of samples were prepared from the produced fermentation broth: a blank group, a fermentation broth degradation group, and a crude enzyme solution degradation group were prepared. In the fermentation broth degradation group, 1.0 μg/mL ZEN was added to the enzyme-containing fermentation broth. The crude enzyme broth degradation group was prepared as follows: the fermentation solution was obtained after filtering the mycelium, 20 mL of the fermentation liquid was added to the ultrafiltration centrifuge tube by using a 3 KDa membrane and then centrifuged at 5500 rpm and 4 °C for 2 h. About 3 mL of the pure protein system and slag filter was obtained. The pure protein system was cleaned 3 times with PBS. In the fermentation broth degradation group and the crude enzyme solution degradation group, 1 μg/mL ZEN standard was added. The blank group, fermentation broth degradation group, and crude enzyme solution degradation group together were mixed in a shaker at 28 °C and 180 rpm for 48 h. After the shaker culture was completed, ZEN with a concentration of 1.0 μg/mL was added to the blank group. After the three groups of samples were extracted, UPLC-q-TOF detection was performed. Each group of samples had 3 biological parallels. Mass spectrometry was conducted under the following conditions:

UPLC-q-TOF was performed using the AB SICEX 5600 system (SCIEX, Framingham, MA, USA). UPLC was performed on a Waters Acquity UPLC system equipped with an ultraviolet detector. The column parameter used was the Waters UPLC HSS T3 column (1.8 μm, 2.1 mm × 100 mm). The column conditions were as follows: column temperature, 35 °C; injection volume, 5 μL; and flow rate, 0.3 mL/min. The mobile phase was as follows: positive mode: A:100% H_2_O (0.1% formic acid); B:100% ACN (0.1% formic acid); negative mode: A:100% H_2_O (0.5 mM NH_4_F); B:100% can; injection volume, 5 μL; and flow rate, 0.3 mL/min. Other parameters were set: capillary voltage, 4.5 kV; cone voltage, 30 kV; source temperature, 120 °C, and desolvation temperature, 350 °C. Mass spectrometry was performed within a scan range of 50 *m*/*z* to 1000 *m*/*z*.

## Figures and Tables

**Figure 1 toxins-13-00720-f001:**
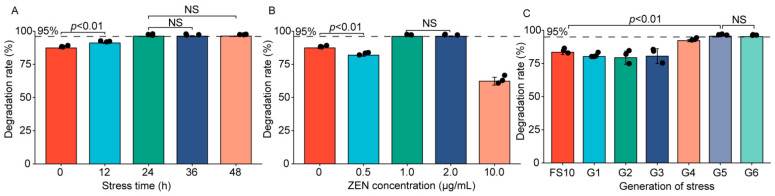
Comparison of degradation rate under different stress conditions and screening of ZEN-S-FS10. (**A**) Comparison of degradation rate under different stress time (stress concentration = 1.0 μg/mL, Number of cycles of stress = 5); (**B**) comparison of degradation rates under different stress concentrations (stress time = 24 h, number of cycles of stress = 5); (**C**) comparison of degradation rates under different stress times (stress time = 24 h, stress concentration = 1.0 μg/mL).

**Figure 2 toxins-13-00720-f002:**
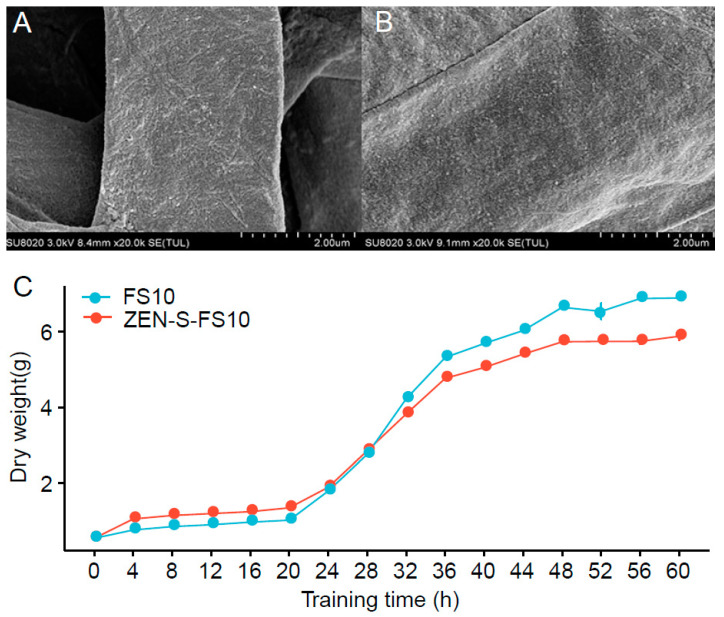
Mycelium characterization and growth ability comparison between ZEN-S-FS10 and FS10. (**A**) SEM image of hyphae of FS10; (**B**) SEM image of mycelium of ZEN-S-FS10; (**C**) growth comparison of FS10 and ZEN-S-FS10.

**Figure 3 toxins-13-00720-f003:**
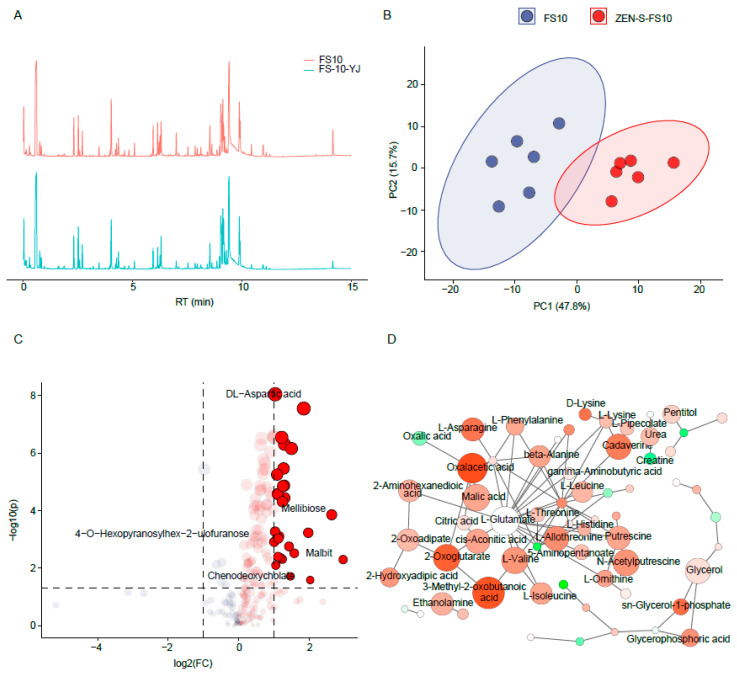
Comparative analysis of mycelium metabolites before and after mutagenesis. (**A**) TIC of FS10 and ZEN-S-FS10; (**B**) PCA; (**C**) analysis of changes in metabolites; (**D**) metabolic pathway analysis.

**Figure 4 toxins-13-00720-f004:**
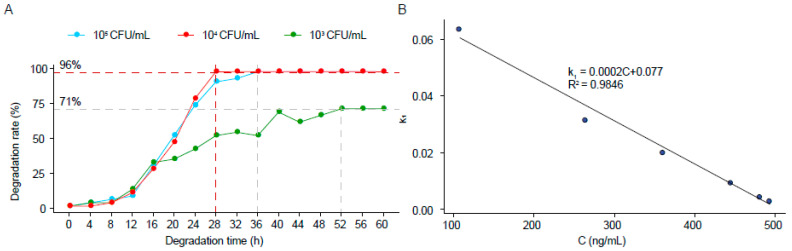
Dynamic degradation curves of ZEN-S-FS10 under different spore concentrations and the fitting results of degradation kinetics: (**A**) dynamic degradation curve of ZEN-S-FS10 under three spore concentrations; (**B**) first-order degradation kinetic model fitting results.

**Figure 5 toxins-13-00720-f005:**
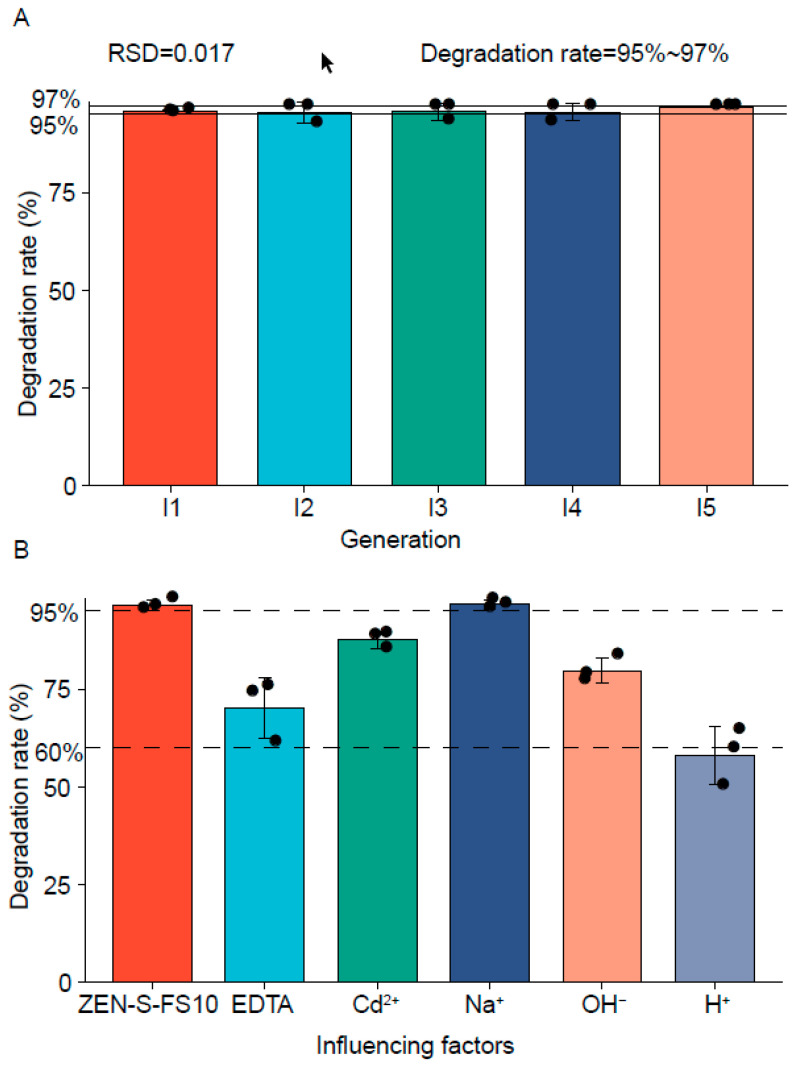
Factors influencing the degradation ability of ZEN-S-FS10. (**A**) After 5 generations, the degradation ability of ZEN was tested. The data were analyzed using the *t*-test. (**B**) Environmental factors influencing the ability of ZEN-S-FS10 to degrade ZEN.

**Figure 6 toxins-13-00720-f006:**
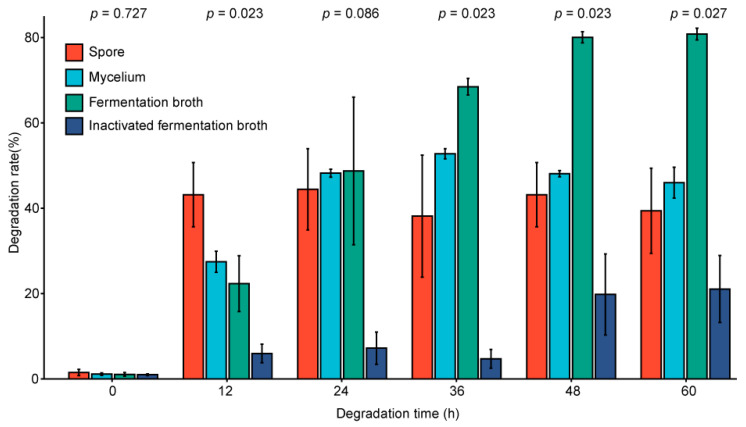
Comparison of degradation ability of different degradation sites under different degradation time.

**Figure 7 toxins-13-00720-f007:**
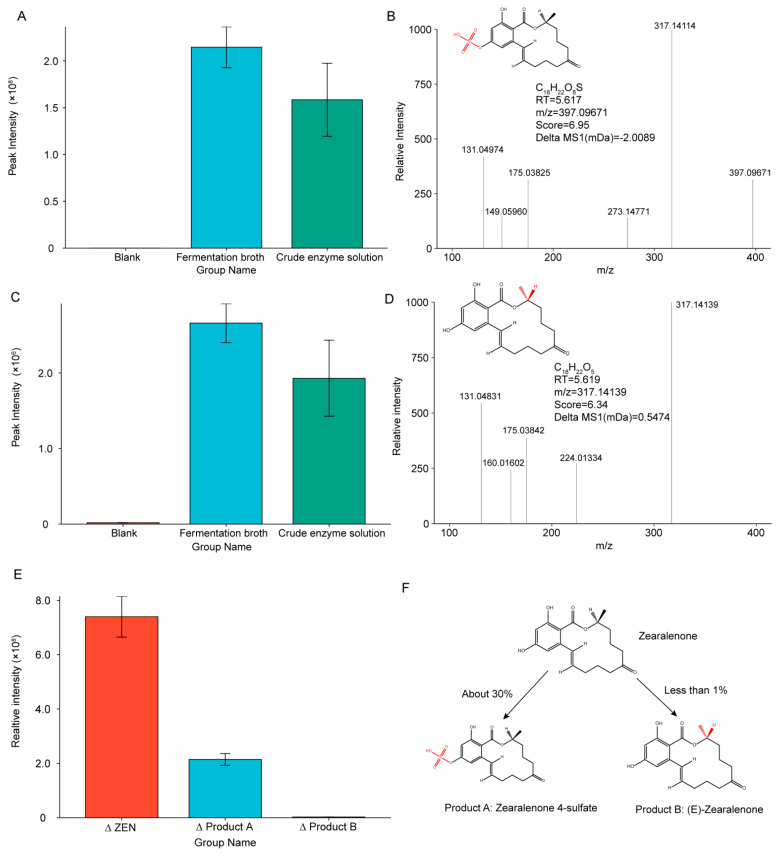
Analysis and structure prediction of ZEN degradation products by high-resolution mass spectrometry UPLC-q-TOF. (**A**) Change in the content of product A in the 3 samples; (**B**) high-resolution mass spectroscopy and structure information speculation of product A; (**C**) change in content of product B in 3 samples; (**D**) high-resolution mass spectrometry and structure information speculation of product B; (**E**) comparison of the change of ZEN with the change of product A and product B; (**F**) fate pathway after ZEN degradation.

**Figure 8 toxins-13-00720-f008:**
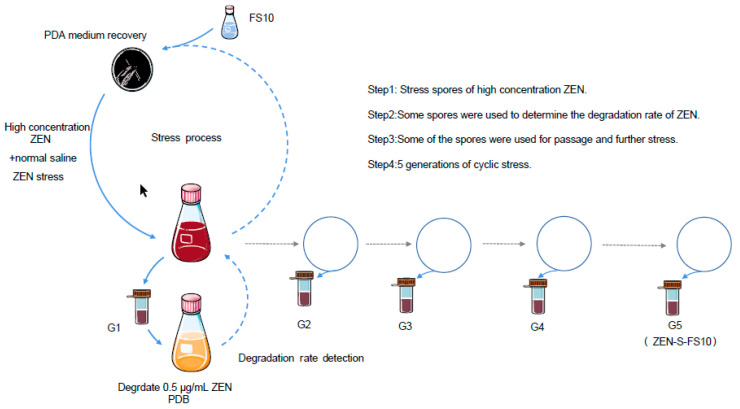
Main process idea and ZEN detection method for obtaining ZEN-S-FS10 after the stress of FS10.

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
