# Peer review of "Exploration on the Enhancement of Detoxification Ability of Zearalenone and Its Degradation Products of *Aspergillus niger* FS10 under Directional Stress of Zearalenone"

_toxins, 2021, doi:10.3390/toxins13100720_

Round 1

Reviewer 1 Report

Line 45: stylistic error

Line 61-63: stylistic error

Line 443: Fig 7 - actually it should be Fig.8

and there is no figure 8....

Author Response

1)Line 45: stylistic error

Response: Thanks for the valuable comments on this article. The format of the reference here has been revised. “ZEN contamination in feed occurs worldwide. Africa has become an area with a high incidence of mycotoxin pollution due to climatic reasons, poor sanitation and epidemic prevention conditions, and lack of sufficient food safety knowledge among the people[8].”, the revised part has been marked in red font, Line 57-60.

  • Line 61-63: stylistic error

Response: Thanks for the valuable comments on this article. We have revised the format of the reference there. “The current methods of biodegradation of mycotoxins mainly originate from four aspects: bacterial degradation (Bacillus subtilis[13], etc.), fungal degradation (Gliocladium rosea[14], etc.), enzyme degradation (ZHD101[15],etc.).” The revised part is marked in red font, Line 76-79.

3)Line 443: Fig 7 - actually it should be Fig.8, and there is no figure 8....

Response: Thanks for the valuable comments on this article. We re-checked and revised the legend. “Fig.8. Analysis and structure prediction of ZEN degradation products by high-resolution mass spectrometry LC-qTOF-MS. (A) Change in the content of product A in the 3 samples; (B) High-resolution mass spectroscopy and structure information speculation of product A; (C) Change in content of product B in 3 samples; (D) High-resolution mass spectrometry and structure information speculation of product B;(E)Comparison of the change of ZEN with the change of product A and product B (F)Fate pathway after ZEN degradation”. The revised part has been marked in red font, Line 503 -509.

Reviewer 2 Report

The work subjected the FS10 strain to stress cycles by ZEN, and after 5 cycles it obtained what the authors call the FS-10-YJ mutation, which obtained a high efficiency of ZEN degradation in vitro (greater than 95%), under optimal degradation conditions (strain concentration of 104 CFU / mL and degradation time of 28 h), also performed pre- and post-stress fungal metabolome analyses, as well as other phenotypic analyses.

The work provides good data on the process of optimizing conditions for ZEN degradation in vitro, however, the authors endeavored to compare the results of FS10 and FS-10-YJ, treating them as wild and mutated individuals, respectively. I don't think it's appropriate to treat the ZEN-stressed FS10 strain as a new individual (FS-10-YJ strain). Even more problematic is the use of the terms “mutation, mutagenesis, generation, mutated”, throughout the text, these terms should be banned as the authors did not provide any evidence at the genomic level that would allow this conclusion.

Furthermore, the phenotypic data presented as differences between FS10 and FS-10-YJ are fragile and can be more parsimoniously explained by the simple fact that the same strain is evaluated under different conditions, stressed and not stressed by ZEN. To impute the differences found in the two situations to the mutation without showing the mutated and wild gene(s) is not plausible.

Another important point is about the identity of FS10, how was it identified as Aspergillus niger?

In Genbank, the only nucleotide sequence available for this strain is from the ITS region (rDNA), which despite being considered the official barcode for fungi, has long been known to be unsuitable for identifying Aspergillus. A simple nBLAST can prove this, since the only deposit for A. niger FS10 ( KX179644) has identical sequence to several species of A. section Nigri. Furthermore, when only type strains are considered, the strain is closer to A. costaricaensis and A. luchuensis than to A. niger sensu stricto. In short, the fungal identity of the A. niger aggregate strains cannot be obtained based on ITS data (which was done by Xu et al., 2013). As recommended by the ICPA (International Commission of Penicillium and Aspergillus), gene sequencing such as CaM and/or RPB2 should be performed. In addition, the A. niger aggregate is a set of cryptic species, the resolution of this group must be based on phylogenetic constructs, and not only on the result of local alignment (BLAST).

If it is confirmed that it is in fact not A. niger, although it is not an error originating in the present work, its publication in this way “Aspergillus niger FS-10-YJ” may contribute to the spread of the error and the contamination of the literature scientific, even affecting the conclusion of the present work (line 466).

In summary, the work has good optimization data and has scientific merit, however, it is structured on an insubstantial premise, treating the same individual as two different ones, and this ends up making much of the discussion unusable and bewilders the work's findings so that its correction would result in a new wording.

Therefore, my recommendation is for not accepting the study for publication in Toxins at this time.

Below there are other details that caught my attention:

In item 2.3, it is not clear how the spore suspension control was carried out, what was the method to reach the control of 106?

Another point, how was the concentration of 0.5 μg/mL of ZEN in PDB medium established?

I don't understand, why was the mycelium drying for electron microscopy? This would only harm the visualization of the structures.

About the sentence: “With an increase  in mutation generation, the degradation rate of ZEN decreases slowly initially and then increases gradually [29].” ...The data presented in Fig.1 do not support this statement, the evolution of degradation rates from G3 to G4 increases abruptly, there is no gradual evolution, and are stabilized in the next “generations”, therefore, gradually it cannot be used in this case.

Lines 241-242: This sentence: ” Moreover, yeast can effectively degrade ZEN in wine” it is completely detached from the context, and does not contribute to the discussion.

Lines 242-247: the sentence is based on reference 27, however, when reading the aforementioned work, I did not find support for such a statement.

The statement :“When fungi gradually adapt to toxin stimulation, they decompose the toxins in the external environment.” ...is based on what reference?

Lines 248-249: why specifically quote the second “generation”?

Line 260: what does reference 31 do here? I didn't find anything related to scanning electron microscopy in the referenced work, it seems to me that the numerical references of the paper are out of order, check please.

About the Lines section 267-276:

The increase in dry weight in relation to the cultivation time and its subsequent stabilization is completely expected. It doesn't seem like a remarkable find to me.

In the sentence: “ We also found that the dry weight of the FS-10-YJ mycelium would eventually be less than that of FS10 when stability was reached” the term “eventually” renders everything else useless, moreover, such a statement must be statistically supported, otherwise it is just speculation.

Author Response

  • The work provides good data on the process of optimizing conditions for ZEN degradation in vitro, however, the authors endeavored to compare the results of FS10 and FS-10-YJ, treating them as wild and mutated individuals, respectively. I don't think it's appropriate to treat the ZEN-stressed FS10 strain as a new individual (FS-10-YJ strain). Even more problematic is the use of the terms “mutation, mutagenesis, generation, mutated”, throughout the text, these terms should be banned as the authors did not provide any evidence at the genomic level that would allow this conclusion.

Response: Thank you very much for your valuable comments. We carefully checked the terms “mutation, mutagenesis, generation, mutated” mentioned in the article. We agree that the ZEN-stressed FS10 strain is not a new individual (FS-10-YJ strain). All the inappropriate expression in the manuscript were revised as stressed strain, as the reviewer suggested, Line 27, 114, 298, 300, 512.

  • Furthermore, the phenotypic data presented as differences between FS10 and FS-10-YJ are fragile and can be more parsimoniously explained by the simple fact that the same strain is evaluated under different conditions, stressed and not stressed by ZEN. To impute the differences found in the two situations to the mutation without showing the mutated and wild gene(s) is not plausible.

Response: Thank you very much for the reviewer's comments. The FS10 stressed with ZEN leads to the change of metabolic level, which is a stable stress change. So, the ZEN- stressed was named as FS-10-YJ. We compared the difference in the metabolic level of FS10 and FS-10-YJ, and showed that the metabolic capacity of FS-10-YJ has been significantly improved, which is mainly manifested in the significant up-regulation of the metabolic pathway of protein synthesis. Based on metabolomics data and evidence of growth and mycelial characterization, ZEN-stressed FS10 (named FS-10-YJ) has a better ability to degrade mycotoxin ZEN.

  • Another important point is about the identity of FS10, how was it identified as Aspergillus niger?

Response: Thank you very much for your valuable comments. We sequenced the 18s rDNA gene of FS-10-YJ and uploaded the results to the NCBI official website for comparison. It has been confirmed that FS-10-YJ belongs to Aspergillus niger. And, we add the results in the supplementary material (Table S2). The modified content has been marked in red font, Line 269-272.

  • In item 2.3, it is not clear how the spore suspension control was carried out, what was the method to reach the control of 106?

Response: Thank you very much for your valuable comments. Sampling the concentration of the spore suspension, observe under a microscope with a hemocytometer, and then count to get its concentration. We have added relevant content to the article, and the modified part has been marked in red, Line 125-129, as follows:

“After sampling the spore suspension, counting on a hemocytometer was used to calculate the spore concentration, and the spore concentration was controlled by the method of diluting with normal saline, and the spore suspension in the PDB medium was controlled to 106 CFU/mL.”

  • Another point, how was the concentration of 0.5 μg/mL of ZEN in PDB medium established?

Response: Thank you very much for your valuable comments. The concentration in the PDB medium was determined to be 0.5 μg/mL by liquid chromatography. We have added relevant content to the article, and the modified part has been marked in red, Line 124-125, as follows:

“The ZEN concentration in the PDB medium was controlled to 0.5 μg/mL after detection by liquid chromatography.”

  • I don't understand, why was the mycelium drying for electron microscopy? This would only harm the visualization of the structures.

Response: Thank you very much for the reviewer's comments. When taking SEM images, the sample must be dried, otherwise the sample cannot be fixed. The pretreatment method of SEM characterization of fungal mycelium, refer to the article of “Shrestha, Bhushan, Sang-Kuk Han, Kwon-Sang Yoon, and Jae-Mo Sung. "Morphological Characteristics of Conidiogenesis in Cordyceps Militaris." Mycobiology 33, no. 2 (2005): 69-76.”

  • About the sentence: “With an increase in mutation generation, the degradation rate of ZEN decreases slowly initially and then increases gradually [29].” ...The data presented in Fig.1 do not support this statement, the evolution of degradation rates from G3 to G4 increases abruptly, there is no gradual evolution, and are stabilized in the next “generations”, therefore, gradually it cannot be used in this case.

Response: Thank you very much for the reviewer's comments. We have carefully modified the content of this place. The description of "gradually increasing" has been revised as: “In addition, as the number of stresses increased, the degradation rate also increased significantly, and finally reached stability at G5 to G6 (Fig.2C).”. The modified content has been marked in red font, Line 264-266.

  • Lines 241-242: This sentence: “Moreover, yeast can effectively degrade ZEN in wine”it is completely detached from the context, and does not contribute to the discussion.

Response: Agree. As this sentence contributes nothing to the discussion, the sentence was deleted.

  • Lines 242-247: the sentence is based on reference 27, however, when reading the aforementioned work, I did not find support for such a statement.

Response: Thank you very much for the valuable comments of the reviewers. We checked the original reference and found it is an insertion error. We have deleted the original reference and replaced with a new one, as follows:

“31. Xiulan, Sun, He Xingxing, Xue Kathy Siyu, Li Yun, Xu Dan, and Qian He. "Biological Detoxification of Zearalenone by Aspergillus Niger Strain Fs10." Food and Chemical Toxicology 72 (2014): 76-82.”

The modified part has been marked in red font, Line 689.

  • The statement:“When fungi gradually adapt to toxin stimulation, they decompose the toxins in the external environment.” ...is based on what reference?

Response: Thank you very much for your valuable comments. We checked the original reference and found it is an insertion error. We have deleted the original reference and replaced with a new one, as follows:

“32. Watanabe, C., I. Terashima, and K. Noguchi. "Responses of the Respiratory Chain to Low Temperature Stress." Plant and Cell Physiology 48 (2007): S181-S81.” The revised part has been marked in red font in the text, Line 692.

  • Lines 248-249: why specifically quote the second“generation”?

Response: Thank you very much for your valuable comments. As the number of stresses increases, degradation rate has increased significantly from G3-G5. And first three generations have no statistical difference. So, we delete the description of second generation in the updated manuscript.

  • Line 260: what does reference 31 do here? I didn't find anything related to scanning electron microscopy in the referenced work, it seems to me that the numerical references of the paper are out of order, check please.

Response: Thank you very much for the valuable comments of the reviewers. We checked the original references and found there has an insertion error of references. All the references have been ordered correctly, in the updated manuscript. Here, we have deleted the original reference and replaced it with a new reference, as follows:

“33. Anggorowati, M. A. "Comparing Pls-Sem and Sem Bayesian for Small Sample in Tam Analysis." International Journal of Applied Mathematics & Statistics 53, no. 5 (2015): 53-58”. The modified part has been marked in red font, line 693.

13)About the Lines section 267-276: The increase in dry weight in relation to the cultivation time and its subsequent stabilization is completely expected. It doesn't seem like a remarkable find to me.

Response: Thank you very much for the valuable comments of the reviewers! The purpose of our dry weight comparison with cultivation time (Fig. 3) is to show how much difference is there in phenotype with and without the stress of ZEN. The results indicated that FS-10-YJ can still obtain a higher degradation rate at a lower growth rate, showing from another angle that its metabolome has changed.

In Fig 6A, the experiment of subsequent stabilization of FS-10-YJ degradation capability is to show that the degradation rate of ZEN-stressed FS10 (FS-10-YJ) is stable and does not decrease with generations.

14)In the sentence: “ We also found that the dry weight of the FS-10-YJ mycelium would eventually be less than that of FS10 when stability was reached” the term “eventually” renders everything else useless, moreover, such a statement must be statistically supported, otherwise it is just speculation.

Response: Thank you very much for the valuable comments of the reviewers! Agree, we also think the expression is not suitable. We revised that sentence as follows:

“We compared the effect of the changes in the hyphae of FS10 and FS-10-YJ as degradation time increased (Fig. 3C). We found that with the increase of the cultivation time, the dry weight of the two myceliums still increased and stabilized with time. We also found that when stability is reached, the dry weight of FS-10-YJ mycelium will be less than the dry weight of FS10.

Reviewer 3 Report

The manuscript provides a well-designed, conducted, and presented study describing the efficient degradation of zearalenone by Aspergillus niger FS-10-YJ. The metabolism of FS-10-YJ and FS10 was compared to prove the differences in metabolites under stimulation of zearalenone. Moreover, the analysis of degradation products and degradation pathways makes this study more interesting.

The manuscript was a pleasure to read and deserves publication.

The authors need to address some minor edits indicated below.

  1. Lines 38 - 41. Replace by a reference of zearalenone
  2. There is a typo in line 273. It should be "weight [34]." instead of "weight. [34]."
  3. There is a typo in line 304. It should be "cells [38, 40]." instead of "cells. [38, 40]"
  4. Lines 443 – 449. Is it Fig.8? I could not find Fig.8 mentioned in section 3.9
  5. I have a small question regarding the degradation rate. Why did the ZEN degradation rate of FS-10-YJ exceed 95% in Fig.6 but lower in Fig. 7 (about 80%)?

Author Response

  • Lines 38 - 41. Replace by a reference of zearalenone

Response: Thanks for the valuable comments on this article. We have deleted the redundant description in the article and replaced it with a reference. “ZEN contamination in feed occurs worldwide. Africa has become an area with a high incidence of mycotoxin pollution due to climatic reasons, poor sanitation and epidemic prevention conditions, and lack of sufficient food safety knowledge among the people[8]”. The revised part has been marked in red font in the text, Line 57-60.

  • There is a typo in line 273. It should be "weight [34]." instead of "weight. [34]."

Response: Thanks for the valuable comments on this article. "weight [36]." has been revised as "weight. [36].". “The reason might be that FS-10-YJ is stimulated by ZEN in the spore state, leading to changes in metabolism, which inhibits the metabolic process related to mycelial growth and cannot maintain the original mycelium weight[36].” The revised part is marked in red font in the text, Line 315.

  • There is a typo in line 304. It should be "cells [38, 40]." instead of "cells. [38, 40]"

Response: Thanks for the valuable comments on this article. "cells [40, 42]." has been revised as "cells. [40, 42]". “Amino acid metabolism can quickly secrete substances that protect cell tissues to reduce the adverse effects of the external adverse environment on cells[40, 42].”. The revised part is marked in red font in the text, Line 348-349.

  • Lines 443 – Is it Fig.8? I could not find Fig.8 mentioned in section 3.9

Response: Thanks for the valuable comments on this article. We re-checked and revised the legend. “Fig.8. Analysis and structure prediction of ZEN degradation products by high-resolution mass spectrometry LC-qTOF-MS. (A) Change in the content of product A in the 3 samples; (B) High-resolution mass spectroscopy and structure information speculation of product A; (C) Change in content of product B in 3 samples; (D) High-resolution mass spectrometry and structure information speculation of product B;(E)Comparison of the change of ZEN with the change of product A and product B (F)Fate pathway after ZEN degradation”. The revised part has been marked in red font, Line 503 -509.

  • I have a small question regarding the degradation rate. Why did the ZEN degradation rate of FS-10-YJ exceed 95% in Fig.6 but lower in Fig. 7 (about 80%)?

Response: Thanks for the valuable comments on this article. What we mentioned in Figure 6 is that the total degradation rate of the spores + hyphae + fermentation broth of Aspergillus niger FS-10-YJ exceeds 95%. In Figure 7, due to the removal of spores and hyphae, only the fermentation broth is used as the main body to degrade ZEN, so the degradation rate has decreased, only about 80%. 

Round 2

Reviewer 2 Report

My considerations remain the same as they have not been adequately answered.
The Results and Discussion section should be rewritten, as it is based on the idea that the FS10-YJ strain is a new individual, the simple exchange of a few words in the text does not change the context of the discussion. Furthermore, the simple naming of the FS10 strain with another name (FS10-YJ) already denotes the idea of ​​a new individual.

Furthermore, from the evidence already mentioned in the previous review, it seems to me that said strain does not belong to the species Aspergillus niger, but to another species of the A. niger aggregate. It is already established in the taxonomy of A. section Nigri that rDNA does not lend itself to identification of this fungal group, there are several species of A. section Nigri that have a sequence of 18s identical to A. niger, so this species cannot be discriminated with based on that locus. The publication of this work in this way (as A. niger) without proper taxonomic analysis may contribute to the propagation of an identification error.

Author Response

1)The Results and Discussion section should be rewritten, as it is based on the idea that the FS10-YJ strain is a new individual, the simple exchange of a few words in the text does not change the context of the discussion. Response: Thanks for the valuable comments on this article. We believe that strains after stress have not undergone major genetic changes, but strains with enhanced metabolic functions after ZEN stress. Therefore, we have rewritten the results and discussion section. The revised part has been marked in red font in the text. The modified parts are as follows: Line 101-111: We used 18s rDNA sequencing method to carry out genome detection of ZEN-S-FS10 and confirmed that ZEN-S-FS10 is still Aspergillus niger strain. The sequencing results are shown in Table S1. In addition, it has reported the key gene RsaI identified by Aspergillus niger[22]. We performed a genetic search on the 18s rDNA sequence of ZEN-S-FS10, the key gene RsaI can be retrieved, and a single band was obtained. From this point, we can think that the metabolism-enhancing strain ZEN-S-FS10 is still a single pure Aspergillus niger. In addition, we detected the peak intensity of the base sequence obtained by sequencing. The result is shown in Fig S1, each base presents a single peak shape with close intensity. This indicates that this strain of Aspergillus niger is a single pure system without other fungal contamination and is not a polymer. Line 407-416: At the same time, we used the gene identification method of Aspergillus niger and other Aspergillus provided by Amaia[22] to sequence the 18s rDNA strain in Wuhan Baiyihui Energy Biotechnology Co., Ltd., and obtained a single band. The sequencing results were posted on the NCBI website ( https://blast.ncbi.nlm.nih.gov/Blast.cgi) to conduct comparative analysis to identify the species of the samples submitted for inspection. At the same time, it was compared with the key genes of Aspergillus niger provided by Amaia[22] to determine the specific species name. In addition, we tested the peak intensity of the base sequence obtained by sequencing to determine the purity of this strain of Aspergillus niger. Line361-365: On the basis of the ZEN degradation efficiency of FS10, we used ZEN to stress the FS10 for 5 times. ZEN-S-FS10, a stressed strain with high ZEN degradation efficiency, was ultimately obtained, with a ZEN degradation rate exceeding 95%. The 18s rDNA and base sequence alignment of this strain confirmed that it is still Aspergillus niger [22] Line 377-385: However, the enzyme class used to degrade ZEN in this strain has yet to be determined. [53]. Moreover, A. niger cannot be directly used in ZEN degradation in food [54]. The aforementioned extracellular degradation enzymes have to be further extracted and purified to develop products with high ZEN degradation efficiency [1, 55]. From this perspective, Metabolism-enhanced FS10 Aspergillus niger ZEN-S-FS10 exhibits research and application potential, which bears significance for ZEN degradation in the future [56, 57]. The research ideas and development methods related to this strain are expected to prompt efforts toward the degradation of other mycotoxins. 2)Furthermore, the simple naming of the FS10 strain with another name (FS10-YJ) already denotes the idea of a new individual. Response: Thanks for the valuable comments on this article. We completely agree with your comments on the modification of name of this strain. We believe that under the ZEN stress, the strain did not form a completely new individual specie. In order to distinguish it from the previous FS10, we need to give a new name to the strain, whose metabolic function enhanced after stress based on our metabolomics result. In previous submissions, the naming of this strain (FS-10-YJ) may lead readers to think that this is a new strain with genetic differences. Therefore, we amended FS-10-YJ in the text to ZEN-S-FS10 (ZEN-stressed-FS 10). The modified part has been marked in red font. 3)Furthermore, from the evidence already mentioned in the previous review, it seems to me that said strain does not belong to the species Aspergillus niger, but to another species of the A. niger aggregate. It is already established in the taxonomy of A. section Nigri that rDNA does not lend itself to identification of this fungal group, there are several species of A. section Nigri that have a sequence of 18s identical to A. niger, so this species cannot be discriminated with based on that locus. The publication of this work in this way (as A. niger) without proper taxonomic analysis may contribute to the propagation of an identification error. Response: Thanks for the valuable comments on this article. We compared the18S rDNA sequencing results of the ZEN stressed Aspergillus niger (ZEN-S-FS10) with that of Aspergillus niger proposed by Amaia et al. 2005(Gonzalez-Salgado, Patino, Vazquez, & Gonzalez-Jaen, 2005). The key gene Rsal, which is unique in Aspergillus niger mentioned by Amaia et al., was found in the gene sequence of ZEN-S-FS10. The gene Rsal could be used for discrimination of Aspergillus niger and other Aspergillus species belonging to section Nigri. The result and sequence are shown in the figure below, which can prove that the strain mentioned in this article is Aspergillus niger. The PCR detection gene sequence of ZEN-S-FS10: TGCCCCCCGGAATACCAGGGGGCGCAATGTGCGTTCAAAGACTCGATGATTCACTGAATTCTGCAATTCACATTAGTTATCGCATTTCGCTGCGTTCTTCATCGATGCCGGAACCAAGAGATCCATTGTTGAAAGTTTTAACTGATTGCATTCAATCAACTCAGACTGCACGCTTTCAGACAGTGTTCGTGTTGGGGTCTCCGGCGGGCACGGGCCCGGGGGGCAGAGGCGCCCCCCCGGCGGCCGACAAGCGGCGGGCCCGCCGAAGCAACAGGGTACAATAGACACGGATGGGAGGTTGGGCCCAAAGGACCCGCACTCGGTAATGATCCTTCCGTTAGGGGAACCTGCGGAAGGATCATTACCGAGTGCGGGTCCTTTGGGCCCAACCTCCCATCCGTGTCTATTGTACCCTGTTGCTTCGGCGGGCCCGCCGCTTGTCGGCCGCCGGGGGGGCGCCTCTGCCCCCCGGGCCCGTGCCCGCCGGAGACCCCAACACGAACACTG In addition, we monitored the purity of the base sequence of ZEN-S-FS10, and the data is provided in Fig S1 in the supplementary material. From the figure, we can see that the peak of each base is a single peak, which indicates that the Aspergillus niger ZEN-S-FS10 is a single pure strain. And we also explained the conclusion in the article, and the rewritten part has been marked in red font, as follows: Line 100-111: From this point, we can think that the metabolism-enhancing strain ZEN-S-FS10 is still a single pure Aspergillus niger. In addition, we detected the peak intensity of the base sequence obtained by sequencing. The result is shown in Fig S1, each base presents a single peak shape with close intensity. This indicates that this strain of Aspergillus niger is a single pure system without other fungal contamination and is not a polymer. Literature comparison results: Fig S1. The content intensity of all base sequences of ZEN-S-FS10 References: Gonzalez-Salgado, A., Patino, B., Vazquez, C., & Gonzalez-Jaen, M. T. (2005). Discrimination of Aspergillus niger and other Aspergillus species belonging to section Nigri by PCR assays. Fems Microbiology Letters, 245(2), 353-361.

This manuscript is a resubmission of an earlier submission. The following is a list of the peer review reports and author responses from that submission.

Round 1

Reviewer 1 Report

1) In the text there were some references to the Tables wich were not added to the text (for example line 328 Table S2)

2) The name of the Figure 4 does not fit with the figure (A, B and C, while there is onle A and B with another content)

3) Line 419 "Fig. S2" ????

Reviewer 2 Report

The article entitled “Exploration on the enhancement of detoxification ability of zearalenone and its degradation products of Aspergillus niger FS-10-YJ under directional stress of zearalenone” reports the capacity of a stressed Aspergillus niger strain to zearalenone degradation. The authors seem to make a deep study of stressed process characteristics and stressed strain characteristics. Although this topic is very interesting and important to safety food issues, the paper need a very deep revision. The theoretical background is insufficient. Several references were randomized (some information cited in this paper is not in accord with the information that appear on the reference document) or not appear on reference list.

I suggest this manuscript should be rejected. Nonetheless, my critical comments are listed below.

Page 1, Line 37 to 39 – It is necessary indicate the source of this information.

All introduction section – Need to improve and clarify information. I think the information put the focus only on China. Mycotoxins is a global issue. I think that is necessary to further explore the “biological degradation”, since your work is about this degradation method.

Page 4, Line 135-136 – Correct the type of letter of “Bis(trimethylsilyl)trifluoroacetamide (BSTFA)”

Page 4, Line 125 – Clarify what is an “EP tube”.

Page 4, Line 139 – Correct the reference “Qui et al.”. This reference does not appear in References section.

Page 8, Line 248 to 249 – This information does not appear in the indicated references.

Page 8, Line 262 – This information does not appear in the indicated references.

Page 10, Line 334 – This is one of your conclusions. Why did you put a reference “S. W. Chen, 334 et al., 2019” in this information?

Page 12, Line 354 to 356 – The reference style can be the same in all document.

Figure 5 – Please, try to improve the quality of the figure. The points appear cut.

Figure 7 – Please, try to improve the quality of the figure.
